# Challenges and Strategies for a Thorough Characterization of Antibody Acidic Charge Variants

**DOI:** 10.3390/bioengineering9110641

**Published:** 2022-11-03

**Authors:** Y. Diana Liu, Lance Cadang, Karenna Bol, Xiao Pan, Katherine Tschudi, Mansour Jazayri, Julien Camperi, David Michels, John Stults, Reed J. Harris, Feng Yang

**Affiliations:** Pharma Technical Development, Genentech/Roche, South San Francisco, CA 94080, USA

**Keywords:** biotherapeutics, antibody, charge variant

## Abstract

Heterogeneity of therapeutic Monoclonal antibody (mAb) drugs are due to protein variants generated during the manufacturing process. These protein variants can be critical quality attributes (CQAs) depending on their potential impact on drug safety and/or efficacy. To identify CQAs and ensure the drug product qualities, a thorough characterization is required but challenging due to the complex structure of biotherapeutics. Past characterization studies for basic and acidic variants revealed that full characterizations were limited to the basic charge variants, while the quantitative measurements of acidic variants left gaps. Consequently, the characterization and quantitation of acidic variants are more challenging. A case study of a therapeutic mAb1 accounted for two-thirds of the enriched acidic variants in the initial characterization study. This led to additional investigations, closing the quantification gaps of mAb1 acidic variants. This work demonstrates that a well-designed study with the right choices of analytical methods can play a key role in characterization studies. Thus, the updated strategies for more complete antibody charge variant characterization are recommended.

## 1. Introduction

Therapeutic proteins, such as recombinant monoclonal antibodies, bispecific antibodies, or antibody fragments, are heterogeneous due to chemical or enzymatic post-translational modifications (PTMs) that occur during their manufacturing process [1,2,3,4]. Many of these PTMs change the overall charge (or surface charge) distribution of the protein, generating charge variants [5,6,7]. The charge variants with a lower intrinsic isoelectric point (pI) than the major constituents are acidic variants, while the ones with a higher pI value are basic variants. Typical acidic variants are species with deamidation, glycation, the sialylated glycan, trisulfide, etc. Basic variants include C-terminal unprocessed lysine, C-terminal amidation, isomerization of aspartate residues, and so on [8,9,10]. Those protein charge variants can be measured and characterized by charge indicating analytical assays, imaged capillary isoelectric focusing (icIEF), ion-exchange chromatography (IEC), or liquid chromatography/mass spectrometry (LC/MS).

Protein variants that affect its immunogenicity, bioactivity, or stability are critical quality attributes (CQAs) [5,11,12,13]. A thorough characterization of the charge variants is important to identify CQAs, which are required by regulatory authorities for therapeutic drug products. The characterization also leads to an improved understanding and guides to establish the process and control strategies via controlling or eliminating the undesired charge variants for better product quality. Various physicochemical assays are utilized to characterize antibody charge variants [12,13]. A conventional analytical chromatography method, IEC, is widely used to separate and isolate protein charge variants. Several MS techniques, such as intact mass analysis and peptide mapping, are followed to identify the exact nature of the modification and its location for isolated charge variants. icIEF is a technique to measure charge heterogeneity of proteins primarily based on a molecule’s pI intrinsic net charge. Size-exclusion chromatography (SEC) is used to detect any aggregates that might be generated during sample preparation for charge variant enrichment. In addition, antibody-based potency assays or cell-based bioassays are used to evaluate drug biological activity, potency, or efficacy. The data from all the assays are combined with structural approaches collectively to establish a thorough understanding of the antibody charge variants and their effect.

Unlike chemically synthesized small molecule drugs, which have a well-defined structure that can be fully characterized, biological drug products are challenging to be fully characterized due to their structural complexity. To illustrate the challenges, we use a recombinant mAb1 as an example. The sequence of this humanized monoclonal antibody immunoglobulin type 1 (IgG1) is based on a human IgG1 kappa (κ) framework, which contains humanized variable (V) heavy chain (HC) region subgroup III (VHIII) and variable light chain (LC) region subgroup I (VκI). Its charge variants are separated using IEC or icIEF. Figure 1 is the ion-exchange chromatography profile of mAb1. The species in the acidic and basic regions were initially characterized and identified using various analytical methods, such as peptide mapping, SEC, CE-SDS, etc. In the basic region, the major basic species were unprocessed lysine (Lys, K), proline (Pro, P) amidation, and signal peptides of LC and HC which were summed up to 93% of total basics. The rest of the basic species can be explained by the oxidation modifications eluting in the basic region. The basic region of mAb1 was fully characterized and the basic species were accounted for about 100%. However, the initial quantitative measurements of identified acidic species left some gaps; all identified acidic species only account for ~65% of total acidic variants (17.2% deamidation, 15.2% glycation, 21.6% oxidation, 0.6% hydroxylation, 10% low molecular weight species (LMWS) and 0.6% Fab glycan). Other possible acidic species, e.g., advanced glycation end-products (AGEs), O-linked glycosylation, and sequence variant (SV), were also investigated and were either undetectable or less than trace level (<0.5%). Note, some attributes were analyzed by peptide mapping thus generating peptide level percentage, 100 × modified/(modified + unmodified), which means percent per chain. The peptide level percentages were multiplied by two to be converted to protein level percentages, as the molecule contains two identical chains. As the result, about one-third of acidic variants remain unidentified. This mystery is not unusual for many biological products [3,14].

The goal of this study was to identify the analysis gap and unaccounted acidic variants in the initial characterization studies. High order structure (HOS) was on the top of the analysis gap list because this long-time mystery led us to speculate the missing acidic variants could be something hard to measure like high order structure. Among a variety of biophysical methods for HOS characterization [15,16,17,18,19,20,21,22], differential scanning calorimetry (DSC) is used to assess potential differences in higher-order structures that would affect thermal stability [17], while hydrogen-deuterium exchange coupled to mass spectrometry (HDX-MS) uses the intrinsic property of amide hydrogens to exchange with hydrogens in solution to track protein conformational changes [18,19].

We also investigated the potential modifications that could have contributed to the missing species in the acidic region and were not studied previously, such as less known lysine succinylation and sialylated glycan. Lysine succinylation is a PTM where a succinyl group (-CO-CH2-CH2-CO2H) is added to a lysine residue of a protein molecule [23,24]. The addition of the succinyl group changes the lysine charge from +1 to −1 at physiological pH and introduces a structural moiety (+100.0186 Da). This modification was rare but was found in proteins, which is expected to lead to more significant changes in protein structure and function [23]. Sialic acids are negatively charged sugar residues present on both N- and O-linked glycan [25,26].

Another possible gap could be the glycation quantification for mAb1. Protein glycation is a common acidic variant, generated due to exposure to glucose in the cell culture medium during antibody production [27,28,29]. Glycation can have profound effects on protein structure, function, and stability [30,31]. Glycation impact on therapeutic antibodies is well studied and the glycation rates in vivo have been assessed [32]. One study found that for a highly glycated mAb with glycation scattered in many sites, the risk to affect potency is low [33]. However, once glycated, proteins can undergo further oxidation of the Amadori compounds to generate a complex set of advanced glycation end (AGE) products, resulting in decreased bioactivity of the protein [34].

The total glycation on a mAb is often measured by boronate affinity chromatography and mass spectrometry-based intact mass analysis [35]. The two methodologies correlated well. Further, the specific sites of glycation are characterized using peptide map methods [33,36]. Ideally, all glycation sites that are detected by peptide mapping add up to provide the overall glycation per antibody chain. However, the glycation measurements from these two assays sometimes do not match. Because peptide mapping often misses low level (below LOD of ~0.2%) glycation that scatters many sites throughout the protein (>80 lysines in mAb1). In addition, the choice of enzymes for peptide mapping makes a difference. The trypsin digestion does not cleave at glycated lysines and often produces larger peptides (i.e., missed cleavages) that pose challenges in the data analysis for accurate glycation quantification. Due to those reasons, the measured level often deviates from the actual level of glycation. A better method to measure total glycation level of the intact molecule is the LC/MS, i.e., intact mass analysis, by which the delta mass +162 Da per glycation can be well resolved from un-modified ones. In addition, LC/MS analysis of the reduced molecule can be performed to identify glycation on LC and HC.

To find the missing acidic variants from the initial charge variant analysis of mAb1, in this extended study we characterized HOS, lysine succinylation, and modifications that do not directly affect charge but showed an increase in acidic regions. We re-examined mAb1glycation measurement using intact mass analysis since the previous study was done by peptide mapping. We also assessed mAb1 sialylation as it could be present in non-consensus Asn glycosylation [37]. It is important to account for all the variants thoroughly. This paper summarizes the lesson learned and discusses protein charge variant characterization strategies that are applicable for other biotherapeutic mAbs.

## 2. Materials and Methods

### 2.1. Material

This study used a recombinant human IgG1 monoclonal antibody with kappa light chain, mAb1, which was produced at the Genentech facility (S. San Francisco, CA, USA). mAb1 was expressed in Chinese hamster ovary (CHO) cells and purified by standard manufacturing procedures. All organic solvents were of analytical or HPLC grade.

### 2.2. IEC Method

IEC was performed using an Agilent 1200 HPLC system (Agilent Technology, Santa Clara, CA, USA). Aliquots of mAb1 were diluted to 10 mg/mL with mobile phase A consisting of 25 mM N-(2-Acetamido)-2-aminoethanesulfonic acid (ACES) buffer, pH 7. The sample (5 µL) was injected for analysis using a strong cation-exchange chromatography column (YMC-BioPro SP-F, 4.6 × 100 mm). The column temperature was at 36 °C. The column was equilibrated with 92% buffer A and 7% buffer B (150 mM Sodium Chloride in mobile phase A) with a 0.6 mL/minute flow rate. Four minutes after 50 µg sample injection, the salt concentration was increased to 15% B over 10 min and then to 20% B at 19 min and 26% B at 22 min. The column was washed with 100% buffer B for three minutes before initial conditions were restored, followed by an 18 min re-equilibration at initial conditions. The total cycle time was 45 min. Absorbance was monitored at 280 nm.

### 2.3. Fraction Collection by IEC

The acidic region and the main peak region were collected by the IEC method as described above with a larger injection (2 mg protein) and multiple injections were required for making enough materials (~2 mg of acidic fraction) for the characterization studies. A one-minute gap between two fractions was taken to avoid cross-contamination. Immediately following the collection, the collected acidic or main fractions were exchanged into the ultrafiltration/diafiltration (UF/DF) buffer to stabilize the protein. The buffer exchange consisted of three times centrifugation (4000× *g*) with an Amicon™ Ultra Centrifugal Filter.

### 2.4. SEC Method

SEC analysis was performed using a Waters H-Class HPLC system. Aliquots of mAb1 fractions were diluted to 1 mg/mL with mobile phase (0.2 M potassium, 0.25 M potassium chloride, pH 6.2). Samples were separated on a TSKgel UP-SW3000 column (Tosoh Bioscience, LLC, South San Francisco, CA, USA) with isocratic elution. The flow rate was at 0.3 mL/min for 18 min, and the column temperature was at ambient. The elution profile was monitored at 280 nm.

### 2.5. icIEF Method

The mAb1 IEC fractions were assessed by icIEF using a ProteinSimple Maurice (San Jose, CA, USA) with a 5 cm-long fluorocarbon-coated capillary cartridge (100 μm ID). The ampholyte solution consisted of a mixture of 0.35% methylcellulose, 2.5 M urea, 0.5% Pharmalyte (GE Healthcare, Chicago, IL, USA) pH 3–10 carrier ampholytes, 2.65% Pharmalyte pH 8–10.5 carrier ampholytes, 0.2% pI marker 7.55, and 0.2% pI marker 9.77 in purified water. The anolyte was 80 mM phosphoric acid, and the catholyte was 100 mM sodium hydroxide, both kept in 0.1% methylcellulose. Fifty microliters of samples were mixed with 200 μL ampholyte solution and then focused at 1.5 kV for 1 min, followed by a potential of 3 kV for 10 min. An image of the focused mAb1 charge variants was obtained by passing 280 nm ultraviolet (UV) light through the capillary and into the lens of a charge-coupled device digital camera. This image was then analyzed to determine the distribution of the charge-related variants.

### 2.6. DSC Analysis

Protein thermal stability was measured using a differential scanning calorimetry, a MicroCal VP-capillary DSC system equipped with an autosampler, and automatic cleaning system (Malvern Panalytical, Malvern Worcestershire, UK). Samples were diluted to approximately 1 mg/mL. A scan rate of 60 °C/h was used over the range of 25–90 °C. The buffer subtraction and baseline were performed in the analysis software. Additionally, the sample concentrations were measured using a microfluidic concentration measurement instrument (Lunatic) and concentration adjustment was also completed in the analysis software.

### 2.7. HDX-MS Analysis

The HDX experiments were performed on an automated LEAP RTC system (Leap Technologies, Morrisville, NC, USA). The samples were diluted to 2 mg/mL by adding 20 mM histidine acetate in H_2_O (pH 5.8). To initiate the HDX, a 4.5 μL aliquot of sample was diluted into 55 μL of 20 mM histidine acetate in D_2_O (pH 5.8) and incubated at 5 °C for different times (36 s to 4 h). The exchange reaction was quenched by adding 55 μL of quenching buffer (8 M urea/1 M TCEP·HCl/0.4 M glycine, pH 2.3). The samples were analyzed by LC/MS using a Dionex UltiMate^TM^ 3000 UHPLC coupled with an Orbitrap Exploris 480 MS (Thermo Fisher Scientific, Sunnyvale, CA, USA). The online digestion was achieved using an immobilized protease XIII/pepsin (1:1) column (2.1 × 30 mm, NovaBioAssays Inc., Woburn, MA, USA) in 0.1% formic acid with 75 μL/min at 10 °C. The proteolytic peptides were collected by a trap column (ACQUITY CSH C18 VanGuard Pre-column, 2.1 × 5 mm, Waters, Milford, MA, USA) and separated by an analytical column (Waters ACQUITY CSH C18 column, 1.0 × 50 mm) at 10 °C with 40 μL/min flow rate. The mobile phases contained 0.1% FA in water (A) and 0.1% FA in ACN (B). The gradient started at 5% B for 1 min and increased to 50% B over 12 min. The protein digests were analyzed by MS with 120 k orbitrap resolution. The peptides were identified using PMi-Byos software (Protein Metrics, Cupertino, CA, USA). The peptide deuterium incorporation was determined by HDExaminer software (Sierra Analytics Inc., Modesto, CA, USA) and ARDD values were calculated as follows [38].
(1)$$\mathrm{ARDD}=\frac{\sum_{i}\frac{A\left(t_{i}\right)-M\left(t_{i}\right)}{M\left(t_{i}\right)}}{i}\times 100\%$$
where A(ti) is the deuterium uptake for the acid fraction at a specific time (ti) and M(ti) is the deuterium uptake for the main fraction at a specific time (ti).

### 2.8. Generating Antibody Fragments Using Enzyme IdeS and IdeE

IdeS (FabRICATOR, Genovis Inc. Cambridge, MA) is a cysteine protease that digests IgG1 at a specific site below the hinge (CPAPELLG/GPSVF), generating F(ab’)2 and Fc/2 fragments. Another cysteine protease IdeE (FabALACTICA, Genovis Inc. Cambridge, MA, USA) digests human IgG1 at one specific site above the hinge (KSCDKT/HTCPPC), generating Fab and Fc fragments. Antibody subunits (Fab, Fc/2, and hinge) were generated using the combination of two enzymes. The digestion procedure is as follows: mAb1 samples were diluted to 1 mg/mL with PBS buffer at pH 7. IdeE protease was added at an enzyme-to-substrate (E/S) ratio of 1:1 (enzyme units/substrate amount in μg) and the mixture was incubated at 37 °C overnight. Then, IdeS protease was added to the mixture (1:1 E/S ratio) for 1-h incubation at 37 °C. Samples were analyzed by LC/MS as described below (Section 2.10).

### 2.9. Desialylation

Samples were diluted to 1 mg/mL with PBS buffer at pH 7. Sialidase (SialEXO, Genovis Inc. Cambridge, MA, USA) was added at a 1:1 E/S ratio (enzyme units/substrate amount in μg) for 2-h incubation at 37 °C. Samples were analyzed by LC/MS.

### 2.10. LC/MS Analysis for Intact Protein and mAb1 Fragments

The fractionated mAb1 samples and appropriate controls were assessed by intact mass analysis. Briefly, the intact sample was diluted to 0.05 mg/mL with MilliQ water and 1 µL was injected into a reversed-phase chip via an Agilent nano-chip system (Agilent Technology, Santa Clara, CA, USA). The sample was desalted and separated in the chip before flow into the mass spectrometer with a 0.1% formic acid in water (A) and 0.1% formic acid in ACN (B) gradient: 0–6 min 25–65% B, 6–8 min 65–95% B, 8–9.5 min 95% B, 9.51–13 min 25% B at 0.4 µL/min. The intact MS data is analyzed using the intact module of the PMi-Byos software (Protein Metrics, Cupertino, CA, USA) using the default parameters with *m*/*z* limited to 2.5 k–3.2 k Da.

### 2.11. Chip Native CE/MS Analysis for Intact Protein

The fractionated mAb and controls were assessed by native CE/MS using a ZipChip CE ion source with autosampler (908 Devices, Boston, MA, USA) coupled with a Thermo Q Exactive UHMR mass spectrometer (Thermo Fisher Scientific, Sunnyvale, CA, USA). Samples were diluted to 1 mg/mL in water and 1 nL was injected into an HR ZipChip with 22 cm separation channel and intact protein background electrolyte (pH 3.2–3.4). CE settings were 0.5 min pressure assist start time with 5 min run time, and 500 V/cm field strength. MS settings were 275 °C capillary temperature, 2 au sheath gas, 200% S-lens RF level, 0 V in-source CID, 150 V IST desolvation, 10 HCD CE, 5 microscans, 25 k mass resolution, 1 × 10^6^ AGC target, 125 ms maximum injection time, and 2 k-6 k Da mass range.

### 2.12. Tryptic Peptide Map Analysis

#### 2.12.1. Peptide Map by LC/MS

The mAb1 charge species in the previous study were analyzed by tryptic peptide mapping. The protein was digested with trypsin (protein: enzyme weight ratio of 40:1) after subjecting to denaturation, reduction and alkylation. The peptide mixture was separated and analyzed using an Acquity (H-Class) UPLC (Waters Corpration, Milford, MA, USA) coupled with a LTQ Orbitrap MS (Thermo Fisher Scientific, Sunnyvale, CA, USA). The protein digest (20 μg) was injected onto an Acquity UPLC Peptide CSH C18 Column (2.1 mm I.D. × 150 mm, 1.7 µm, 130 Å) was used with 77 °C column temperature. Mobile phases A and B consisted of 0.1% TFA in water (A) and 0.08% TFA in acetonitrile (B). After holding at 0% B for 2 min, a linear gradient was run from 0% to 20% B in 15 min, to 40% in 30 min, and 55% in 15 min, followed by a wash step 95% B for 4 min before column re-equilibrium at the initial condition (7.5 min). The flow rate was set to 0.2 mL/min. Xcalibur™ software was used for the data analysis.

#### 2.12.2. mD LC/MS Analysis

The mAb1 charge species were separated by CEX, fragmented by on-column tryptic digestion, and analyzed using LC/MS. The mD LC/MS method conditions were captured in a previous study [39]. A bio-compatible Dionex UltiMate™ 3000 Rapid Separation mD LC system was coupled to a Q-Exactive mass spectrometer with heated electrospray ionization (HESI) ion source (Thermo Fisher Scientific, Sunnyvale, CA, USA). The Poroszyme™ immobilized trypsin cartridge (2.1 mm I.D. × 30 mm, Thermo Fisher Scientific, Sunnyvale, CA, USA) was employed. The LC acquisition was controlled by Chromeleon™ software, whereas the Q Exactive™ was controlled by Xcalibur™.

## 3. Results

### 3.1. Purity and Quality Check of Collected Acidic and Main Fractions

Acidic and main peak mAb1 fractions collected by the cation exchange analytical column were re-injected for IEC purity analysis along with the unfractionated starting material (Figure 2). Concentrations were determined by comparing total peak areas to the starting material. The purities were 98.2% for the acidic fraction and 97.7% for the main peak fraction.

The fraction collection and buffer exchange process might generate undesired aggregates for the collected materials; we performed quality checks of collected fractions by size exclusion chromatography. Lower levels of high molecular weight species were observed by size exclusion chromatography in the collected acidic (0.1%) and main peak (0.2%) fractions compared to the starting material (1.1%).

### 3.2. Charge Variant Determination by icIEF Method

icIEF assay measures pure charge variation without the structural influences that may occur with IEC. Figure 3 shows the icIEF profiles and the relative peak areas of IEC collected acidic and main peak fractions. The IEC acidic fraction showed 90.8% acidic variants by icIEF. The purity differences (90.8% vs. 98.2%) of acidic fractions by icIEF and IEC can be explained by the different separation mechanisms and resolution of the two methods for charge variants [29]. IEC separates antibody populations primarily based on the surface charge of solvent-exposed residues combined with hydrophobic interactions. The slightly lower purity by icIEF is unsurprising, as IEC elution positions can also have hydrophobic or other structural factors [5]. Therefore, the ~10% of IEC separated charge variants were likely due to the structural variants that cause surface-charge changes of solvent-exposed residues via hydrophobic interaction change.

### 3.3. HOS Studies by DSC and HDX-MS

DSC is used to characterize the stability of a protein directly in its native form. It measures the heat change associated with the molecule’s thermal denaturation when heated [40]. HDX-MS is a rapidly evolving technique for analyzing structural features and dynamic properties of proteins [41]. Both methods were used to unveil possible HOS differences of mAb1 acidic and main fractions.

#### 3.3.1. DSC Analysis

As basic terms of DSC analysis measure the midpoint unfolding temperature (T_m_) that is the temperature at the peak maxima of the endothermic transitions. T_m_ values are useful for elucidating the impact on protein HOS changes, and therefore relating conformational changes to potential functional impact. In this investigation, we used DSC analysis for the mAb1 IEC fractions and the thermograms were recorded across the thermal ramp as shown in Figure 4. The similar 86 °C transitions and smaller 59 °C transitions for both acidic and main fractions indicate that no structural differences can be detected by DSC.

#### 3.3.2. HDX-MS

The HDX time course was designed with five different time points from 36 s to 4 h. The protein samples were digested online via an immobilized protease column and analyzed by LC/MS. The percentage of average relative deuterium-uptake difference (ARDD) was generated to assess the difference in deuterium uptake levels between the peptides in the two fractions. For comparison, Figure 5 overlays the deuterium-uptake time courses for selected peptides of the two IEC fractions. The results revealed no difference between the acidic and main fractions. For all the peptides of mAb1, the ARDD values were less than 5% (−2.1% to 3.2%). Previous studies indicated that the ARDD values greater than 5% represented the HOS difference [42,43]. Therefore, no difference in HOS was detected for the two IEC fractions by HDX-MS analysis.

### 3.4. LC/MS Analysis of Additional Potential Acidic Charge Variants

Several LC/MS analyses were conducted for the acidic and main fractions: intact antibody, antibody fragments, and protease digests, to investigate possible charge contributing covalent modifications such as sialylated glycan, lysine succinylation, and modifications that were enriched in the acidic region such as pyroglutamate formation, that were not studied previously.

#### 3.4.1. Sialylated Glycan

To determine whether sialylated glycans were present in the molecule, sialidase digestion was used to remove sialic acids from mAb1 if any. The mAb1 samples with and without sialidase digestion were analyzed by LC/MS. No difference was observed for both samples, indicating no detectable sialylated glycan. For confirmation, mAb1 was digested into Fab and Fc fragments, using IdeS and IdeE enzymes; to utilize the better sensitivity for lower level variants obtained with lower molecular weights. The consistent results for antibody fragments with or without sialidase treatment confirmed that the acidic region does contain sialylated glycans.

#### 3.4.2. Lysine Succinylation

This modification was searched using multi-dimensional (mD) LC/MS analysis of trypsin-digested mAb1. No lysine succinylation, indicated by non-cleavage and an increase of 100 Da [23], was found in the acidic fraction.

#### 3.4.3. Pyroglutamate

Cyclization of N-terminal glutamic acid (Glu, E) into pyroglutamate (pyro-Glu, pE) is a familiar PTM of proteins [44]. With no net charge change related to the conversion from Glu to pyro-Glu, the pyro-Glu variants can elute at different regions by IEC due to surface charge differences via local conformational changes. Pyro-Glu by tryptic mapping was found at 3.6% in the acidic fraction and 2.8% in the main fraction, indicating only slightly higher pyro-Glu in the acidic fraction than in the main peak fraction.

### 3.5. Glycation Quantification by Intact Mass Analysis

Glycation with delta mass +162.0528 Da can be analyzed using LC/MS or capillary electrophoresis–mass spectrometry (CE/MS) methods. We measured glycation at the protein level for IEC acidic and main fractions by intact mass analysis in this study; the percentages of protein glycation were calculated based on the integrated peak intensity areas of deconvoluted mass spectra shown in Figure 6. The glycation percentages of acidic and main fractions were 37.4% and 18.3%, respectively, much higher than the initial analysis (15.2% for the acidic region and 11.2% for the main peak) results by tryptic map analysis. CE-MS results confirmed the higher glycation levels obtained for intact LC/MS analysis. This indicates the peptide map method under-estimated glycation, which may be due in part to the dimeric nature of mAbs reducing our ability to see low-level variants against a background of unmodified structure.

## 4. Discussion

### 4.1. Closing the Gaps of mAb1 Acidic Variant Quantification

We conducted this study to close gaps in our understanding of the acidic variants of a recombinant monoclonal antibody. Protein charge variants are commonly observed for recombinant mAbs, and their product quality impacts are highly dependent on the modified sites, types, and levels. While identifying the site provides insights for criticality assessment of the modification, the modification level is important to evaluate product quality effects. For example, the different effects of deamidation on different antibodies could be due to their interactions with different antigens, but the percentage of deamidation most likely also plays a role [45,46]. Therefore, it is essential to characterize thoroughly both the protein charge variant modification site(s) and levels. In this study, the isolated IEC acidic fraction was subjected to additional characterization studies to fill in the missing acidic variant gaps from previous studies. This study excludes HOS differences, lysine succinylation and sialylation as the undetected acidic form. The result revealed unaccounted for glycation and pyro-Glu in the acidic fraction; the major unaccounted for acidic modification was glycation. Overall, as shown in Table 1, the identified acidic variants now account for ~91% of the IEC acidic region from ~65% initially. The ~ 9% acidic variant remaining unknown at this time might be due to structural variants that cause surface-charge changes detected by IEC, or other variants that cannot be characterized or are underestimated by the analytical tools we explored.

Although both peptide mapping and intact mass analysis can detect protein glycation, only the total glycation level measured by intact mass analysis with the intact molecule correlated well with the percentage of total acidic peaks measured by IEC [47]. Trypsin and lysyl endopeptidase (Lys-C) do not cleave at glycated lysine sites, therefore, producing longer peptides, which poses challenges in the data analysis for accurate glycation quantification. The recommended method for glycation quantification is intact mass analysis. Note that mAb1 is not glycosylated in the Fc region, but a low-level non-consensus Fab glycosylation was found, which does not interfere with the glycation measurement. Thus no deglycosylation was needed for the sample prior to the intact mass analysis. For more common mAbs with N-glycans, enzymatic deglycosylation (such as PNGase F) is recommended to remove the glycans that complicate the quantitation especially given that hexose moieties including galactose add the same incremental 162 Da mass as a glycated lysine residue. Other acidic variants that can be detected using intact LC/MS methods but might be overlooked when an isolated acidic fraction is characterized solely using peptide mapping, e.g., glucuronidation and half-antibody fragments.

### 4.2. Recommend Strategies for Characterization of Protein Charge Variants

Based on the lessons learned in this study, the updated strategies for protein charge variant characterization are summarized into a workflow, as shown in Figure 7. Fraction collection is the first step toward a thorough characterization. With the advance in recent years, online mD LC/MS has been used more often and is a good technology for protein characterization [39]. The mD LC/MS analysis, with many immobilized columns available commercially, is automated and fast by saving the collection process. Offline fraction collection is still needed to obtain a relatively larger amount of materials for other important assays such as bioactivity assays to determine the function effectively.

Unstressed drug substance as the material for protein variant characterization is recommended, as the acidic or basic species in the stressed sample might not have the same modification composition. Regional fractions instead of individual peaks should be collected, since the control strategy including reported IEC values is often linked to regional categories rather than individual peaks. This practice also enables early bioactivity assessment. Collection and characterization of individual peak fractions are usually only needed when there is an observed loss of biological activity (such as potency) or a novel attribute with a regional fraction.

It is critical to minimize the artifacts that are generated during fraction collections and sample preparations for protein variants characterization. For example, aggregation, oxidation, or deamidation can occur during sample preparations due to exposure to various pH, agitation, buffer exchange, enzymatic digestion, etc. Some labile modifications (e.g., succinimide) can also be lost during sample preparation. To reduce the artifacts, we immediately exchanged the fraction buffer to ultrafiltration and diafiltration buffer following the collection to stabilize the molecule. In addition, it is always a good practice to include the main peak as a control and carry out side-by-side characterization, which was what we did in this study.

All collected regional fractions should be first evaluated by IEC, icIEF, and variable pathlength extension (SoloVPE) assays to get information on the purity and the concentration, and to ensure their quality prior to further characterization studies. Next, different characterization assays are performed to determine the attributes indicated under the assays in Figure 7. Attributes in green are acidic; the purple-colored are the basic species. The black coloured attributes are neutral in net-charge, could elute in basic, main, or acidic regions, depending on the molecule and its local structure environment. Activity assays help understand if charge variants can have a potential efficacy impact. HOS assays can be used case-by-case for further investigation. The basic fraction is subject to an IEC with and without carboxypeptidase B (CpB) to determine the overall level of C-terminal unprocessed lysine [48]. Some assay choices (whether or not conducted) depend on the molecule studied., e.g., glycan assay is only for the molecules with glycans. Table 2 and Table 3 listed primary assays recommended for the variant quantatition that have been observed in acidic and basic IEC regions, respectively. Note many attributes can be detected by multiple analytical methods, The recommended assays in the tables are the ones that provide good quantification results based on our experience.

## 5. Conclusions

It is essential to perform functional and physico-chemical characterization to identify CQAs that need to be controlled. This mAb1 case study highlights the importance of starting with an inquiry into what to measure and how to measure protein variation. In this study, determining glycation at the protein level using intact mass analysis instead of peptide mapping provided the most accurate information. The majority of the mAb1 acidic variants (~91%) have been identified and accounted for interdependent modification events. The analytical characterization strategies as described here should be generally applicable for other biotherapeutic mAbs.

## Figures and Tables

**Figure 1 bioengineering-09-00641-f001:**
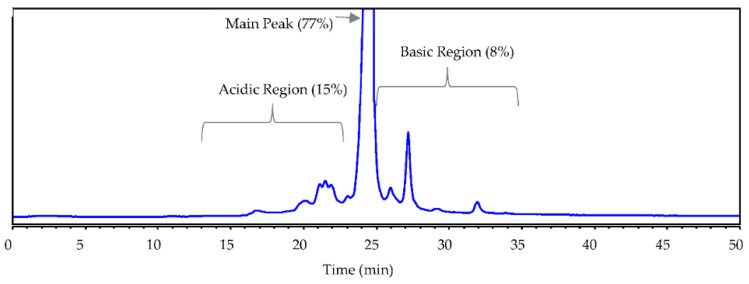
The mAb1 IEC profile by cation exchange chromatography at pH 7 with a salt gradient.

**Figure 2 bioengineering-09-00641-f002:**
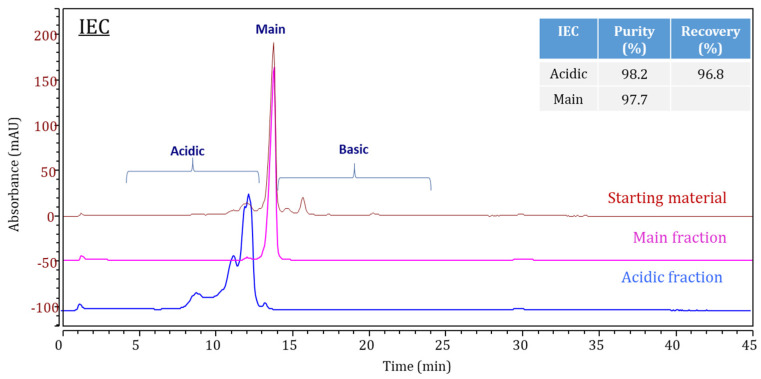
IEC Chromatogram overlays of mAb1 starting material (**mango**), the acidic fraction (**blue**), and the main fraction (**pink**). An inserted table listed the purity and recovery for the acidic and the main fractions.

**Figure 3 bioengineering-09-00641-f003:**
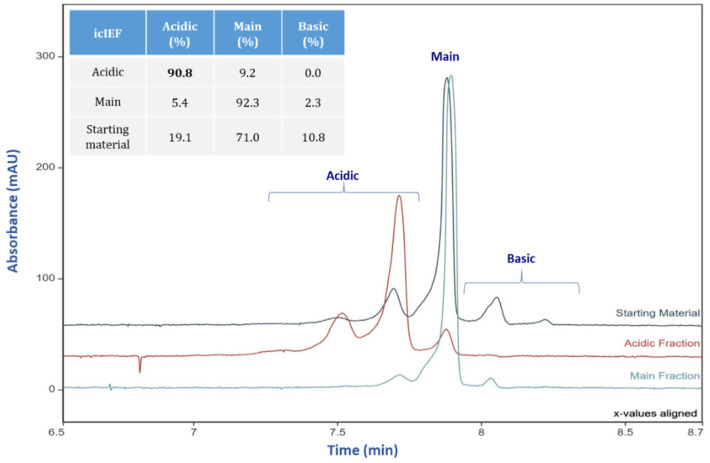
icIEF Electropherogram overlays of mAb starting material (**teal**), the acidic fraction (**mango**), and the main fraction (**blue**). An inserted table listed the levels of acidic, main and basic species measured by icIEF.

**Figure 4 bioengineering-09-00641-f004:**
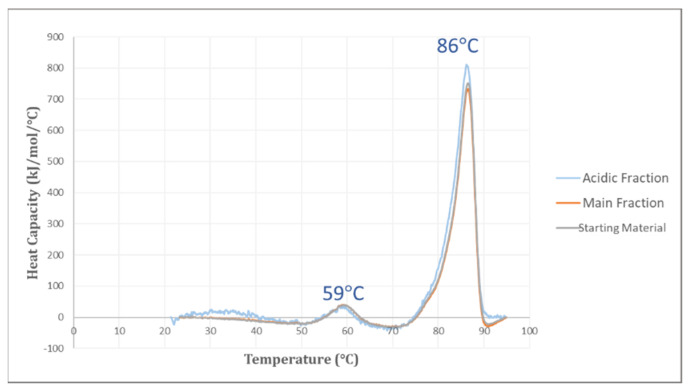
DSC Thermograms of the mAb1 acidic fraction (**blue**), the main fraction (**mango**), and the starting material (**grey**).

**Figure 5 bioengineering-09-00641-f005:**
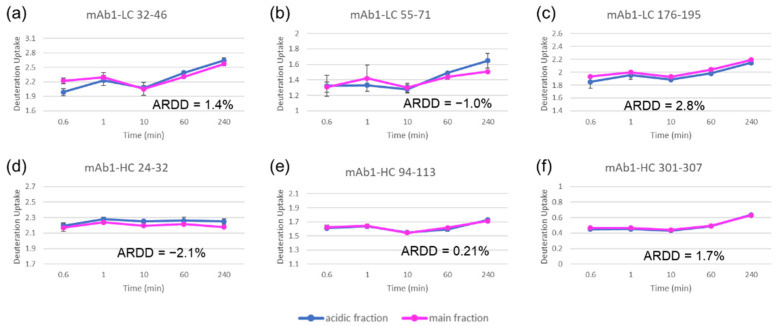
HDX-MS deuterium uptake time course plots for selected mAb1peptides of mAb1 Acidic (blue) and Main (pink) Fractions with the ARDD value for each peptide: mAb1-LC 32–46 (**a**), mAb1-LC, 55–71 (**b**), mAb1-LC 176–195 (**c**), mAb1-HC 24–32 (**d**), mAb1-HC 94–113 (**e**), and mAb1-HC 301–307 (**f**).

**Figure 6 bioengineering-09-00641-f006:**
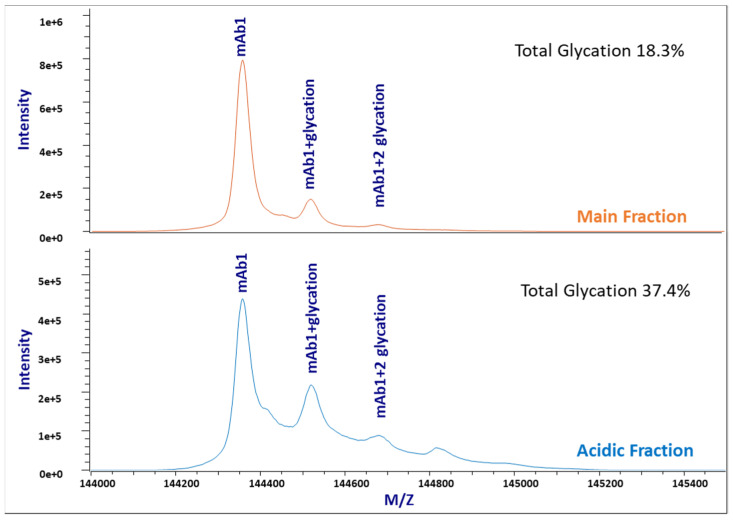
Stacked intact mass analysis TIC plots of the mAb1 main fraction (**top**) and the acidic fraction (**bottom**), showing different glycation levels.

**Figure 7 bioengineering-09-00641-f007:**
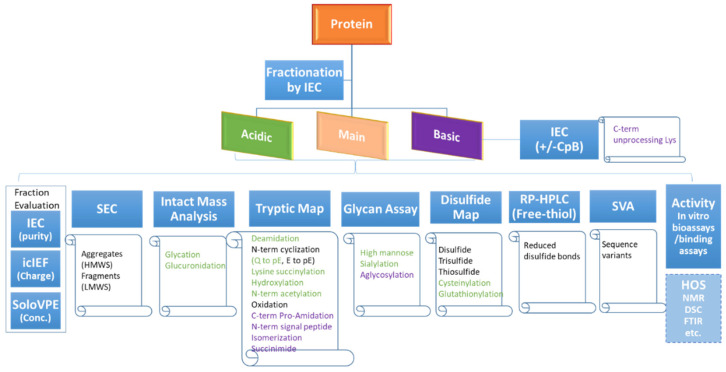
A flowchart diagram showing the charge variant characterization workflow with recommended assays to monitor various product attributes.

**Table 1 bioengineering-09-00641-t001:** Summary of Identified Acidic Variants in mAb1.

Attributes	Percentage	Assays
Deamidation	17.2	
Oxidation	21.6	
Hydroxylation	0.6	Peptide Mapping
Fab Glycan	0.6	
Lys Succinylation	ND *	
Pyroglutamate	3.6	
LMWS	10.0	SEC
Glycation	37.4	Intact mass analysis
AGE	ND *	
O-glycan	ND *	
HOS	ND *	DSC, HDX-MS
**Total identified species**	91.0	

(ND * = not detected).

**Table 2 bioengineering-09-00641-t002:** Recommended Assays to Monitor and Quantify Acidic Variants.

Modifications Observed in Acidic IEC Species	Assay
Deamidation	Tryptic map
N-terminal cyclization (Q to pE)	Tryptic map
Modification by maleuric acid	Tryptic map
Lysine succinylation	Tryptic map
Hydroxylation	Tryptic map
Sialylation	Glycan assay
High mannose	Glycan assay
Glycation	Intact mass analysis
Glucuronidation	Intact mass analysis
Fragments (LMWS)	SEC
Cysteinylation	Non-reducing Lys-C map
Glutathionylation	Non-reducing Lys-C map
Disulfide isoforms	Non-reducing Lys-C map
Trisulfide	Non-reducing Lys-C map
Reduced disulfide bonds (free-thiol)	RP-HPLC with NcME-tagging
Oxidation	Tryptic map
N-terminal cyclization (E to pE)	Tryptic map

**Table 3 bioengineering-09-00641-t003:** Recommended Assays to Monitor and Quantify Basic Variants.

Modifications Observed in IEC Basic Species	Assay
C-terminal un-processing Lys	IEC with/without cpb
C-terminal Pro amidation	Tryptic map
N-terminal signal peptide	Tryptic map
Isomerization	Tryptic map
Aggregates (HMWS)	SEC
Glycosylation	Glycan assay
Succinimide	Tryptic map
Oxidation	Tryptic map
Fragments (LMWS)	SEC

## Data Availability

Not applicable.

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
