# Peer review of "Challenges and Strategies for a Thorough Characterization of Antibody Acidic Charge Variants"

_bioengineering, 2022, doi:10.3390/bioengineering9110641_

Round 1

Reviewer 1 Report

This is an interesting article on strategies for characterising acidic forms of monoclonal antibodies. It is an important approach for the production of these biopharmaceuticals and is a very worthwhile article to publish.

Have only minor considerations to make:

1. I have identified in the text a number of spelling or typing errors that will certainly be corrected by the final version of the text, but I ask for attention;

2. The last sentence of the introduction (lines 126 to 128) is a little vague. A simple reformulation would be enough to make it clearer and more coherent with the rest of the text (which, by the way, is very well written);

3. Lines 131-132 - I suggest a better description of the antibody, as done in the introduction.

Reviewer 2 Report

This study of Liu et al is technically well performed and provide interesting data for the characterization of antibody acidic charge variant. Anyone involved in the pharmaceutical development of antibodies knows how difficult it is to characterize the post-translational modifications (PTM) that occur during their manufacturing process. PTMs may impact the stability, PK/PD and binding-activity of the manufactured product.

The paper is well written and read well. The authors focus on a case study which is a humanized IgG1 produced by CHO cells which are considered as the workhorse for therapeutic nacked whole antibodies production. First, they report a step by step isolation and an accurate characterization of acidic variants by using various and complementary analytic metods. They propose a strategy and a workflow with recommended assays to monitor various product attributes and quantify basic variants. This will be of great interest during the pharmaceutical development of a drug candidate. I do appreciate the merit of the authors work and the general interest from a drug developer point of view. I have no major criticisms on their work, only minor points to reconsider and suggestion that could potentially contribute to improve the discussion

Minor Comments: 

Line 60 (introduction) refers to figure 1 for the first time. Therefore, Legend 1 should be more accurate and should indicate the type of IEC (cation exchange), the pH and the gradient type. In addition, it would also be interesting to compare in an insert the peak areas (Acidic versus Main versus Basic).

Size-exclusion chromatography is commonly used to detect any aggregates that might be generated during sample preparation for charge variant enrichment, but no data are shown here. is there any reason for that ?

Typo:

line 53 "Unlike" and not "Uunlike"

line 446 "neutral" and not "neural"
